# Experiences of participants in a clinical trial of a novel radioactive treatment for advanced prostate cancer: A nested, qualitative longitudinal study

**Bianca Viljoen**[1,2], **Michael S. Hofman**[3,4], **Suzanne K. Chambers**[2,5,6,7], **Jeff Dunn**[2,6], **Haryana M. Dhillon**[8], **Ian D. Davis**[9,10], **Nicholas Ralph**[1,2] *

1 School of Nursing & Midwifery, University of Southern Queensland, Toowoomba, Queensland, Australia, 2 Centre for Health Research, University of Southern Queensland, Springfield, Queensland, Australia, 3 Prostate Cancer Theranostics and Imaging Centre of Excellence (ProsTIC), Molecular Imaging and Therapeutic Nuclear Medicine, Peter MacCallum Cancer Centre, Melbourne, Australia, 4 Sir Peter MacCallum Department of Medicine, The University of Melbourne, Melbourne, Victoria, Australia, 5 Faculty of Health Sciences, Australian Catholic University, Brisbane, Queensland, Australia, 6 Prostate Cancer Foundation of Australia, Sydney, New South Wales, Australia, 7 Exercise Medicine Research Institute, Edith Cowan University, Perth, Western Australia, Australia, 8 Centre for Medical Psychology and Evidence-based Decision-making (CeMPED), School of Psychology, Faculty of Science, The University of Sydney, Sydney, Victoria, Australia, 9 Eastern Health Clinical School, Monash University, Melbourne, Victoria, Australia, 10 Department of Medical Oncology, Eastern Health, Melbourne, Victoria, Australia

* nicholas.ralph@usq.edu.au

## Abstract

### Objectives

Qualitative studies nested within clinical trials can provide insight into the treatment experience, how this evolves over time and where improved supportive care is required. The purpose of this qualitative study is to describe the lived experiences of men with advanced prostate cancer participating in the TheraP trial; a randomised trial of $^{177}$Lu-PSMA-617 compared with cabazitaxel chemotherapy.

### Methods

Fifteen men with advanced prostate cancer were recruited from the TheraP clinical trial with interviews conducted at three timepoints during the trial. An interpretative phenomenological approach was used, and interviews analysed using thematic analysis. This research paper reports the results from the mid-point, conclusion and follow up interviews, focusing specifically on participants' experiences of trial participation.

### Results

Three themes were identified representing the lived experiences of men with advanced prostate cancer participating in the TheraP trial: (1) facing limited options; (2) anticipating outcomes and (3) coping with health changes.

**Data Availability Statement:** All relevant data are within the paper and its Supporting Information files. We were not given ethical approval to release

all of the data due to it containing sensitive patient information. This restriction is imposed by the Peter MacCallum Human Research Ethics Committee (HREC/48397/PMCC-2018). For data access, please contact the Peter MacCallum Human Research Ethics Committee (ethics@petermac.org) or University of Southern Queensland Human Ethics Committee (human. ethics@usq.edu.au).

**Funding:** Funding support was received from a ANZUP Below the Belt Grant (QualTheraP: a nested, multi-perspective longitudinal qualitative study of participants) (MSH, SKC, JD, HMD, IDD, NR). The National Health and Medical Research Council (NHMRC) Centre of Research Excellence provided further funding support through a PhD Scholarship (BV). The funders had no role in study design, data collection and analysis, decision to publish, or preparation of the manuscript.

**Competing interests:** The authors declare there are no competing interests.

## Conclusions

Men who enrol in clinical trial of anti-neoplastic treatments for prostate cancer need targeted psychological and supportive care that includes attention to unique aspects of the experience of having prostate cancer and being in a clinical trial. As part of their trial experience, men with advanced prostate cancer need to be regularly assessed for survivorship needs, fully informed, supported and referred to services for regular care and support across the trajectory of their disease.

## Trial registration

NCT03392428. Registered on 8 January 2018 (ANZUP1603).

## Introduction

Clinical trials are essential in advancing health care by evaluating safety, efficacy and effectiveness of new health care interventions [1]. For men with advanced prostate cancer the therapeutic landscape has significantly changed over the last decade with the availability of new treatments and increased clinical trial options [2]. A new and effective class of therapy for men with advanced prostate cancer is Lutetium-177 [$^{177}$Lu] Lu-PSMA-617. TheraP (ANZUP 1603; clinicaltrials.gov identifier NCT03392428) was the first open label, randomised phase two clinical trial comparing the effects of Lutetium-177 PSMA-617, a novel radionuclide therapy, to the current standard of care chemotherapy cabazitaxel, in men with progressive metastatic castration resistant prostate cancer [3]. This unique treatment involved a novel dual approach of imaging and treatment, theranostics, in which one radioactive drug was used to identify and provide a graphic representation of the cancer using a PET scan. Then a second radioactive substance $^{177}$Lu-PSMA-617 delivered therapy to treat the main tumour and metastatic tumours [3]. 291 men were screened, in which 200 were eligible to participate in the TheraP trial. Study treatment was received by 98 men randomly assigned to $^{177}$Lu-PSMA-617 and 85 randomly assigned to cabazitaxel [3].

The primary aim of the study was to compare the effects of the two treatments on participants PSA levels [3]. The results of the TheraP trial included a decrease in PSA levels by 50% or more. For men randomly allocated to $^{177}$Lu-PSMA-617 this occurred in 66% and 37% in those assigned to cabazitaxel [3]. PSA rise or progression, revealed on CT or bone scans, indicated that those who received $^{177}$Lu-PSMA-617 were 37% less likely to progress. When reviewed at 12 months 19% receiving $^{177}$Lu-PSMA-617 had not progressed, contrasted to 3% in cabazitaxel [3]. Furthermore, tumour shrinkage occurred in 49% of men allocated to $^{177}$Lu-PSMA-617 compared to 24% with cabazitaxel. Significantly, $^{177}$Lu-PSMA-617 caused less severe adverse effects (grade 3–4 toxicities) at 33% than cabazitaxel at 54%. $^{177}$Lu-PSMA-617 is now recognised as a potential alternative treatment to cabazitaxel [3].

The unique context of clinical trial participation among men with advanced prostate cancer elicits similarly unique needs [4]. Men with advanced prostate cancer are known to experience higher psychological distress and risk of suicide, poorer quality of life, additional health and financial burdens and supportive care needs compared to men with localised disease [5, 6]. For men with advanced prostate cancer the advantages of clinical trial participation can include close observation of their condition and support by specialist oncology staff [7], a pathway to improved health [8] and enhance overall survival [7]. New treatments for men with

advanced prostate cancer need to be closely monitored and researched in order to extend life without increasing health-related burdens [4].

Prospective qualitative research can facilitate better insight into the dynamic illness and treatment experience of individuals and identify how these experiences, quality of life, and needs, change over time [9]. Research into the long-term experiences of men with advanced prostate cancer has been limited [10]. The only study to date longitudinally monitored men's outcomes after diagnosis of advanced prostate cancer for five years. These men received androgen deprivation therapy or radiation therapy [10]. Key results were quantified, revealing that forty-six percent of men were highly distressed at diagnosis and thirty-three percent remained so five years later [10].

Another study conducted in Australia qualitatively examined the lived experiences of thirty-nine men with advanced prostate cancer primarily treated with androgen deprivation therapy, radiation therapy or prostatectomy [6]. The experiences of these men detailed regret about late diagnosis and treatment decision, being discounted in the health system, fear/uncertainty about the future, acceptance of their situation, masculinity and treatment effects [6]. A key gap in the literature still remains; examining the lived experiences of men with advanced prostate cancer enrolled in a clinical trial. There is a need to better understand the experiences and corresponding supportive care needs of men with advanced prostate cancer participating in a clinical trial in order to improve the overall care they receive [11].

Accordingly, we undertook a prospective nested, longitudinal qualitative study describing the lived experiences of men with advanced prostate cancer enrolled in the TheraP trial. In a preceding, baseline paper [4] we reported participants motivations, perceptions and experiences of deciding to enrol in the TheraP trial. The results identified four themes including hoping to survive, needing to feel informed, choosing to participate and being randomised [4]. For men with advanced prostate cancer, the experience of deciding to enrol in a clinical trial is occupied with uncertainty, emotional complexities and focused on a desire to survive [4]. The decision to enrol in a clinical trial is mainly determined by a desire to live but also a need to make an informed decision [4].

In this paper we prospectively describe the lived experiences of these men with advanced prostate cancer participating in the TheraP clinical trial [12] and how this changed over time.

## Materials and methods

### Study design

For this prospective, longitudinal, nested qualitative study, we used an interpretative phenomenological approach as it facilitates detailed examinations of personal lived experiences and allows researchers insight into how research participants make sense of a given phenomenon in a complex and emotionally laden context [13].

### Participants

Fifteen men with advanced prostate cancer participating in the TheraP clinical trial were recruited to partake in this qualitative study, following ethical approval from the Peter MacCallum Human Research Ethics Committee (HREC/48397/PMCC-2018). Participants were recruited from the Royal Brisbane and Women's Hospital Brisbane and Peter MacCallum Cancer Centre in Melbourne. Eligibility criteria for TheraP trial recruitment and QualTheraP are discussed in the preceding baseline paper [4].

We interviewed participants at three time points in order to better understand their experiences of the trial as it progressed. Baseline results (time point 1) are reported elsewhere [4], These time points included after enrolment but pre-randomisation (commencement of trial),

at mid-point (11 weeks) and conclusion (23 weeks) of treatment, with a follow up interview after trial participation ceased.

## Procedure

This analysis focused on prospectively describing the lived experience of men with advanced prostate cancer participating in the TheraP clinical trial, therefore it utilised data from interviews conducted at the mid-point (11 weeks), conclusion (23 weeks) of treatment, and a follow up interview after trial participation ceased. Fifteen men were initially interviewed after enrolment but pre-randomisation (commencement of trial). At mid-point (11 weeks) of treatment, three participants were deceased. The final number of participants is represented in Fig 1.

All participants were contacted by researchers (BV, NR) and interviewed via telephone. This form of communication was considered most appropriate for the participant group. Consent was confirmed verbally at the beginning of each interview. Semi-structured interview protocols were used (Table 1) allowing participants to openly express their experiences of trial participation including side effects, changes in quality of life and supportive care needs. Interviews lasted 30 minutes to 1-hour and were audio-recorded, transcribed, and de-identified. Two participants withdrew for personal reasons, one was removed due to the onset of a medical condition, and another placed into palliative care. Participant demographic data is shown in Table 2.

## Data analysis

The transcripts were analysed using thematic analysis with results presented as narrative synthesis. Researchers (BV, NR) independently examined transcripts for common themes and experiences. The approach to thematic analysis by Braun and Clarke [14] was used by coders. Interview transcripts were coded and categorised using NVivo V.12 [15] by (BV). Two additional members of the study team (SC, JD) evaluated theme categorisation to improve reliability and validity of the data. Reporting of data conforms with the Criteria for Reporting Qualitative Studies (COREQ) guidelines [16].

## Results

Three overarching themes were identified signifying the lived experiences of men with advanced prostate cancer participating in the TheraP clinical trial: ***(1)Facing limited options; (2) Anticipating outcomes;*** and ***(3) Coping with health changes***. These findings extend baseline (pre-randomisation) analysis which reported experiences related to: *(1) Hoping to survive; (2) Needing to feel informed; (3) Choosing to participate;* and *(4) Being randomised.* To represent the identified themes and participant's perspectives including similarities and differences, selected quotations from interview transcriptions are shown in Table 3.

## Facing limited options

Knowing that the treatment they received depended on the process of randomisation, participants from both the LuPSMA and cabazitaxel arms, acknowledged that even though that may not obtain their preferred treatment choice, they would still receive appropriate treatment. Although most men reported a desire to receive the experimental treatment (LuPSMA), as reported at baseline [4] and across the timepoints of the trial, participants acknowledged the difference of a trial compared to usual care in that their choice was to participate in a trial rather than to decide on a treatment.

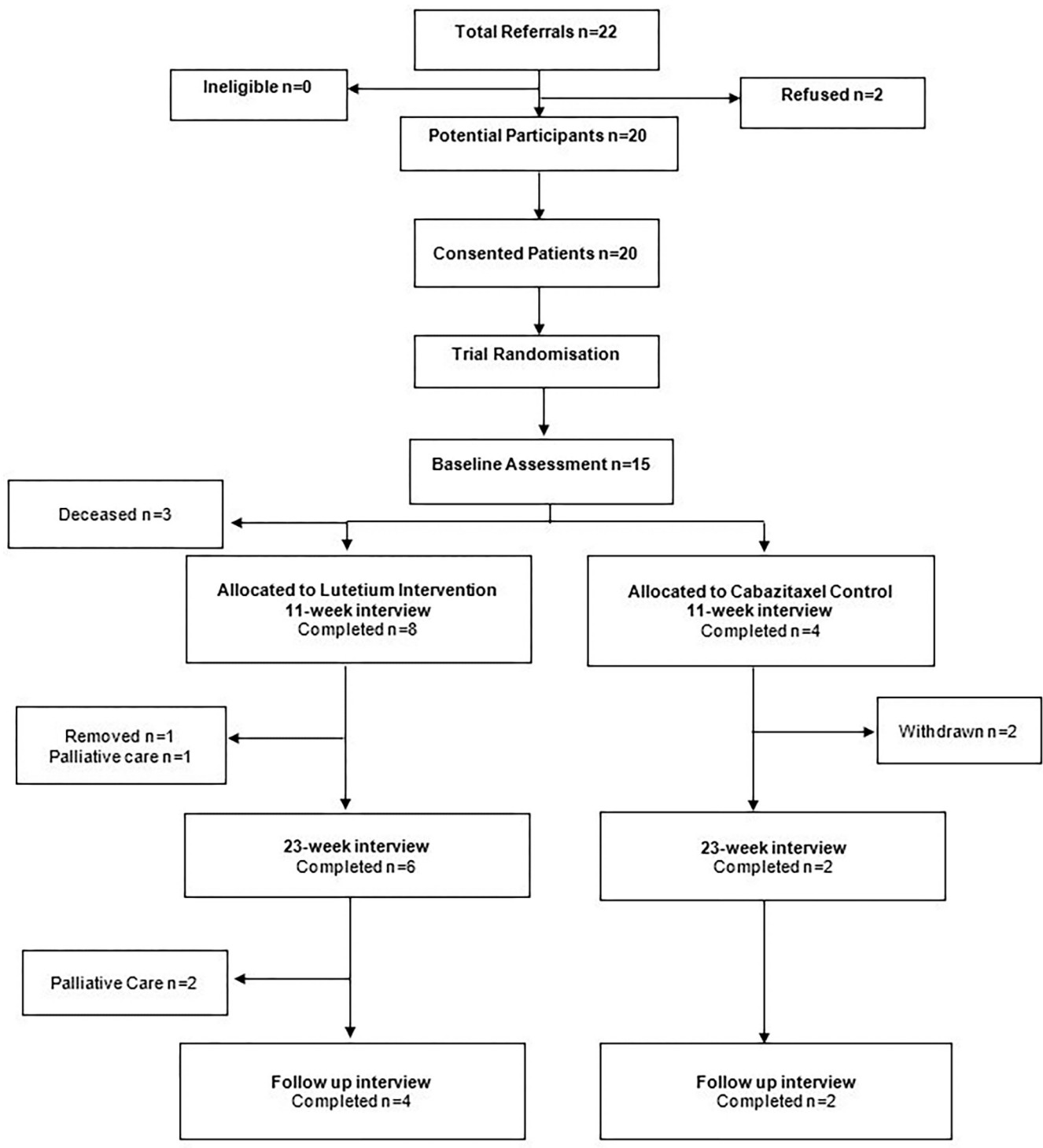

**Fig 1. Study flow diagram.**

**Table 1. Participant interview protocol.**

| Interview 2 Mid-Point (11 Weeks) Questions |
| --- |
| 1. What is your experience of the particular treatment you receive? |
| a. What has been helpful? |
| h:\22287020\uploadb. What has been unhelpful or difficult? |
| 2. How has your participation in the trial affected you physically and psychologically? |
| a. Quality-of-life changes? |
| b. Emotionally or from a coping point of view? |
| c. Financial or practical? |
| 3. In which situations would you have liked more support or care during the trial? |
| a. What would that have looked like? |
| 4. Thinking back what were your initial motivations for participating in this trial? |
| a. In what ways has this changed over time? |
| **Interview 3 Conclusion of Trial (23 Weeks) Questions** |
| 1. Reflecting on the last few months, what would you say has been your personal experience of participating in this trial? |
| a. What has been helpful? |
| b. What has been unhelpful or difficult? |
| 2. How has your participation in the trial affected you physically? |
| a. Quality of life changes? |
| b. Emotionally or from a coping point of view? |
| c. Financial or practical? |
| 3. In which situations would you have liked more support or care during the trial? |
| a. What would that have looked like? |
| 4. Thinking back what were your initial motivations for participating in this trial? |
| a. In what ways has this changed over time? |
| **Interview 4 Follow Up Questions** |
| 1. Reflecting on the last few months, what has been your personal experience since the trial concluded? |
| a. Quality of life changes? |
| b. Emotionally or from a coping point of view? |
| c. Financial or practical? |
| 2. Have any plans been established, treatment or otherwise, going forward? |

*I'm not saying it [LuPSMA] was going to cure me, but I thought I was going to be had relief for cancer for after I'd even finished you know, a year or 2 years"* [ID009, LuPSMA]

*"Being randomised to the Cabazitaxel, again I got disappointment, because I approached them to go onto the lutetium. . . because it, well it gave a bit of an extension of time."* [ID004, cabazitaxel]

Being randomly allocated to either LuPSMA or cabazitaxel reflected the limited treatment options within the clinical trial and elicited responses ranging from contentment they received their preferred treatment to resignation they did not. These perceptions emerged in most participants after the randomisation process was complete. One participant, summed up the perspective of most participants, stating:

*"You just put yourself in there and you've got a 50% chance of getting a different one [treatment]. Whichever way it was going to fall, if I got in, I was going to accept and just go down that road because either way I need treatment."* [ID011, LuPSMA]

Some participants allocated to the cabazitaxel arm reported privately seeking and paying for LuPSMA treatment. This was not identified by those allocated to the experimental treatment.

## Anticipating outcomes

Following allocation to either cabazitaxel or LuPSMA, participants reported engaging in a process of supposition and anticipation regarding the result of treatment. Anticipating outcomes

**Table 2. Participant demographics.**

| | | Mean, SD | Range |
|---|---|---|---|
| | **Age** | 72.38 | 53–85 |
| | **Years since Diagnosis** | 9.14 | 3–20 |
| Age of Participants in larger TheraP Trial (intention to treat population) | Lutetium | 71.7 | |
| | Cabazitaxel | 71.5 | |
| **Variable** | **Number\*** | **Variable** | **Number** |
| **TheraP Treatment Arm** | | **Marital Status** | |
| Lutetium (total n = 99) | 10 | Single | 3 |
| Cabazitaxel (total n = 101) | 5 | Married | 9 |
| **Residency** | | Widowed | 1 |
| Queensland | 8 | Divorced | 1 |
| Victoria | 6 | **Previous Treatment** | |
| **Education** | | Hormone Therapy | 14 |
| Low Level | 4 | Chemotherapy | 12 |
| Medium Level | 3 | Radiation Therapy | 8 |
| High Level | 4 | Radical Prostatectomy | 4 |
| **Occupation** | | | |
| Employed | 1 | | |
| Unemployed | 1 | | |
| Retired | 10 | | |

\*Missing data due to attrition and/or participants not returning the demographic questionnaire

\*Previous treatment includes multiple received by participants

ranging from one where participants presumed an outcome (mostly that LuPSMA would be effective in relieving their symptoms), expecting a poorer outcome from cabazitaxel, wondering whether the allocated treatment would extend their life, speculating about possible treatment side-effects, and subsequently planning for treatment after the trial. Participants also interpreted their allocation as an influential factor in the ultimate trajectory of their disease and their survival:

*"I looked at the list of side effects, and I suppose the one that's going to affect me is the chemo"* [ID001, LuPSMA]

*"I recognise that the disease is progressing and there's a trajectory, a known outcome. I'm hoping to make it [life] last longer."* [ID004, cabazitaxel]

Throughout the trial, participants anticipated trial outcomes based on their individual response to treatment. For instance, on receiving treatment, participants in the LuPSMA treatment arm in particular reported continuously monitoring their treatment response by evaluating the test results, their feelings of physical health, changes in biomarkers of their disease, and the subsequent impact of these changes on the mental challenges they experienced.

*"I'll have a talk to [treating physician] and he sits me down and shows what it was and how it is you know, the last scan and this scan, and he said he can see that the cancer cells are fading out, they're fading out, so it must be successful."* [ID010, LuPSMA]

*"The PSA is coming down, so it seems to be achieving the desired result."* [ID012, LuPSMA]

**Table 3. Themes, subthemes and representative quotations.**

| THEMES | QUOTES |
|---|---|
| *Facing limited choices* | *"I was given some sheets to read over about the possibilities. It wasn't, where I could choose, it was randomly selected"* [ID008, Cabazitaxel] |
| | *"Whichever way it was going to fall, if I got in, I was going to accept and just go down that road because either way I need treatment."* [ID011, Cabazitaxel] |
| *Anticipating outcomes* | **LuPSMA** |
| | *"My way of thinking is if I have the Cabazitaxel there is a good chance I won't have the Lutetium down the track, but if I have the Lutetium and it doesn't work at least I've got the Cabazitaxel as a backup for my next line of chemo".* [ID003] |
| | **Cabazitaxel** |
| | *"Being randomised to the Cabazitaxel, again I got disappointment, because I approached them to go onto the lutetium... because it, well it gave a bit of an extension of time."* [ID004] |
| | *"So, when I had the randomisation results, I thought, oh, you know, maybe that's going to shorten its trajectory, maybe not."* [ID004] |
| *Coping with health changes* | **Physical changes** |
| | **LuPSMA** |
| | *"I shouldn't say pleasant but it's a hell of a lot better than the standard chemo process that's for sure."* [ID003] |
| | *"It's just the fatigue. I haven't got my bone pain, or anything like that, which is really good...an extensive reduction in pain."* [ID006] |
| | *"People tell me I'm looking well, so you know seeing kind of has got to be believing isn't it."* [ID012] |
| | *"I don't feel bad as in, yeah like, I'm functioning quite well."* [ID011] |
| | *"I think I've been so lucky to have truly because the side effects aren't there like on chemo. And I'd love other people to experience what I've experienced. So, if anything it's justified. If I could...be a part of making a drug available to a lot of other people that wouldn't have been without the trial. Then it's worth it."* [ID011] |
| | *"I'm really, really, really pleased, I'm really pleased I got onto the program, and it's done me–I'm actually feeling really, a hundred percent."* [ID009] |
| | **Cabazitaxel** |
| | *"I feel too weak at the moment, to even drive. I'm not prepared to drive the car. So, that is on hold."* [ID008] |
| | *"I suppose as long as they're happy with the decrease in activity of the cancers, then I'll continue on, but I know I was in with another fellow and he's nearly done, completely clear and I'm still quite significant."* [ID011] |
| | **Mental changes** |
| | *"At the moment I'm trying to remain positive and I'm hoping that there are still other things that–that can be done for me."* [ID014, Cabazitaxel] |
| | *"I get a little bit anxious now about you know, what the next phase of treatment will be for me, now that trial has completely finished."* [ID014, Cabazitaxel] |
| | *"I don't really know how long after you stop taking the Lutetium, you're going to be well for...I dare say everybody's different."* [ID009, Cabazitaxel] |
| | *"I don't suppose it's ever going to last forever but that's what the trial's all about, to see how long it's going to last and what side effects."* [ID002, Cabazitaxel] |
| | *"The chemo has been doing its job, now I'm getting to the stage where I'm afraid to wonder how much longer the chemo will be effective."* [ID014, Cabazitaxel] |
| | **Practical Challenges** |
| | *"I'm not working, you just can't, it's getting a bit harder, to cover all these costs."* [ID011] |
| | *"I mean we have to stay overnight because we live away and I'm radioactive and travelling into the city...is hard...so it all adds up."* [ID011] |

*"All the scans and everything that you get exposed to by being in the trial, all the reviewing and stuff that happens about you, that's been really good because you keep an eye on cancers that are not responding, so it just makes me feel as ease."* [ID011, LuPSMA]

Based on physical and mental changes experienced from treatment, participants began to assign their treatment success to the treatment received within the trial by either the way they felt changes, or the way change was measured, or both. Participants reported the results of ongoing scans and diagnostic tests provided clarity regarding their health status, the effectiveness of treatments and explanation for whether a treatment was working. Participants expressed feelings of treatment success primarily by restored functionality and ability to connect socially with family and friends:

*"I've always been, sort of a strong person, and I used to pick up things, really heavy and all that kind of thing. I couldn't do that when I wasn't very well, because. . .it hurts your arms or muscles or whatever. But now, I reckon I'm back to square one again."* [ID009, LuPSMA]

### Coping with health changes

The experience of coping with changes related to alterations in *physical health* and *mental health*, and the *practical challenges* associated with the requirements of trial participation and burdens of cancer and its treatment.

**Coping with changes to physical health.** For participants who received LuPSMA, mostly positive views were expressed about the way they physically responded to treatment with particular emphasis on the limited side-effects and reduction in pain, although fatigue was experienced by most. All patients in the TheraP study had received prior docetaxel chemotherapy, so patients were able to relate side effects from LuPSMA to prior experience with chemotherapy.

*"The big difference I noticed is it's not as intrusive as other chemo's. . .with the Lutetium, I've had none of that [side-effects]. I shouldn't say pleasant but it's a hell of a lot better than the standard chemo process that's for sure. It's just the fatigue. I haven't got my bone pain, or anything like that, which is really good. . .an extensive reduction in pain."* [ID003, LuPSMA]

For individuals who received cabazitaxel, mostly negative views were reported about their physical response with reports centring on debilitating fatigue, lack of physical function, and the "cure" being worse than the disease:

*"I don't know whether the cure is worse than the disease. . .the side effects are just not worth it you know."* [ID012, cabazitaxel]

*"I am normally a very active person, so I think it has had some impact on–on what I've been able to do. I've certainly been able to get about and walkabout and carry on with life but maybe not as much in energy as I have previously."* [ID014, cabazitaxel]

**Coping with changes to mental health.** Participants emphasised that the experience of participating in a trial required an ongoing attitude of resilience. Participants reported the physical changes experienced, both due to the effects of underlying prostate cancer and treatment, resulted in changes to their mental health. In men who felt unwell, this resulted in feelings of worry and anxiety that their treatment was not effective:

*"I think it's a case of, like anything, you know, you have your days when you're a little bit down and you have days where you're back up, you know, but you've got to stay strong and fight all the way through otherwise, it means giving up."* [ID003, LuPSMA]

The side-effect profile of both treatments was also seen to impact on mental status as participants reported that they experienced fewer side-effects associated with LuPSMA and therefore were less anxious about receiving treatment.

*"Definitely not as big of an impact [on mental health], because you're not worried about the treatment knocking you around. So, you don't tend to get a bit anxious when the time comes [for treatment]. That in itself makes you feel better. . .you just haven't got all those problems that come with chemo."* [ID011, LuPSMA]

For individuals who received cabazitaxel, they reported being encouraged by the effectiveness of the treatment in slowing or halting their disease progression yet were still hoping for treatments in addition to that which they were receiving:

*"I'm still feeling pretty positive, but things are, you know, my illness is stable which is encouraging. Obviously if it starts to take a decline again than maybe, mentally, I might feel a bit different."* [ID014, cabazitaxel]

Participants indicated a range of ongoing support required about how long any anti-neoplastic effects of treatments would last and what subsequent care they should receive with many expressing a fear of disease progression given their advanced disease. Notably, participants expressed a range of negative emotions, including worry and anxiety, associated with concerns regarding the follow-up care they would receive following trial participation:

*"I think the time will come even perhaps before the end of the trial when you know, in conjunction with the consultant, we have to think about maybe a plan B going forward."* [ID014, cabazitaxel]

**Practical circumstances.** Throughout the trial, a range of practical challenges and ongoing support needs related to changing circumstances emerged. Many participants reported noticing their practical circumstances were changing due to the impact of their disease, the effects of anti-cancer treatment, and the associated commitment of participating in a clinical trial practical challenges with the impact of the trial on their life in terms of the financial cost, time commitment, and travel requirement:

*"You're just putting the k's [kilometres] on the car and you get the car serviced, the cost of hospital parking. . .it all adds up. . .you just notice it's a price you pay for being on the trial."* [ID011, LuPSMA]

*"I try to keep life pretty uncomplicated, but the trial requires a lot of time and effort and life sort of goes on hold until it's done."* [ID009, LuPSMA]

## Discussion

The participant experiences in our nested study of men in the TheraP trial with advanced prostate cancer who received either [177]Lu-PSMA-617 or cabazitaxel were represented by facing

*limited options*; *anticipating outcomes*; and *coping with health changes* in physical and mental health and practical circumstances. Overall, findings indicate that men who enrol in a clinical trial of anti-neoplastic treatments for prostate cancer need targeted psychological and supportive care that includes attention to the unique aspects of the experience of having advanced prostate cancer and being in a clinical trial.

Firstly, there is a need for improved information-based support with our findings indicating that participants favoured the experimental treatment over the control group. Our participants expressed a strong preference towards [177]Lu-PSMA-617 both at baseline [4] and throughout the trial despite it being an untested treatment and notwithstanding evidence for the effectiveness of cabazitaxel as a treatment for men with advanced prostate cancer [17, 18]. Negative perceptions about the prospect of chemotherapy treatment are reported in the literature [19]. Reasons for such favourable views of the experimental treatment instead of chemotherapy are due to either previous experience or identified side effects within literature, including nausea, vomiting, hair loss, loss of appetite and fatigue [20]. Importantly though, the fact that participants reported hope amidst uncertainty suggests they viewed their disease as curable with the prospect of curative treatment associated with the presence of hope in a trial a reported in a recent review of reviews. Additionally, participant expectations were consistently high at all time points when discussing the potential of [177]Lu-PSMA-617 as an effective treatment for their condition irrespective of their allocation. Although the hypothesis of the TheraP trial reflected clinical equipoise by assuming that there was no good basis to favour [177]Lu-PSMA-617 over cabazitaxel, it may be that reports of the "game-changing" potential of [177]Lu-PSMA-617 both in the scientific literature [21–23] and main stream media [24] may have driven high participant expectations of the experimental treatment in our sample. Additionally, the availability to access LuPSMA treatment privately and the possibility of treated men sharing their survivorship experiences on peer-to-peer online platforms suggest a need for clinicians and researchers to pay closer attention to trial participants' expectations of treatment and provide psycho-educational support that addresses information needs, supports shared decision-making, and truly informed consent. This issue was highlighted in the phase 3 VISION trial of [177]Lu-PSMA-617 [25] where there was a high rate of trial discontinuation in the control arm, requiring implementation of enhanced trial-site education measures.

Participants reported favourable experiences associated with trial participation such as enhanced therapeutic relationships with health professionals and better insight into their disease, progression and response to treatment. For example, men in this study reported persistent satisfaction with their condition being more intensively monitored as part of the trial with frequent scans, tests and consultations with health professionals making them feel more in control of their disease, even when their disease was progressing. Despite conditions that might otherwise suggest more negative experiences such as advanced disease, limited choices, and uncertain treatment outcomes, participants still drew satisfaction and feelings of being in control from a trial context that facilitated an improved insight into their individual response to treatment and improved dialogue with clinicians. These findings mirror study results which report increased satisfaction with cancer care when treatment is personalised, and shared-decision making is facilitated [26]. Nevertheless, the requirements of a trial can often help shape such favourable patient experiences although further research is needed to articulate the nature of this dynamic for it to be repeated in both trial-based supportive care initiatives and broader survivorship care.

In the context of *limited choice* and *anticipating outcomes*, men reported a diverse range of challenges that resulted in their experience as marked by *coping with changes* to their physical and mental health and practical needs associated with their disease, treatment, and trial participation. Given the breadth of physical and mental burden faced by men with prostate cancer,

health professionals working in trials need to develop improved strategies to personalise and support the treatment decisions of men with prostate cancer. Importantly, ensuring tailored referral of men with higher supportive care needs is critical. However, there are no within-trial supportive care interventions reported in the literature for men with advanced prostate cancer in trials and more broadly, there is a lack of interventions for prostate cancer survivors (30). Crawford-Williams et al. [27] state that there are substantial knowledge gaps in prostate cancer survivorship research, reducing the capability of improving men's outcomes across the survivorship experience. More specifically the gap in knowledge of survivorship care for people with metastatic disease has been discussed [28], and more recently a Prostate Cancer Survivorship Essentials Framework has been developed to facilitate this [29–31].

We propose men be comprehensively and routinely assessed for survivorship needs on trial enrolment, fully informed of the short and long-term implications of experimental and control treatment(s); supported in developing their own within-trial goals of care; and referred to services for regular care and support across the course of their disease as part of their trial experience.

## Strengths and limitations

Due to the longitudinal assessment, a significant strength of this study is that it provided information about participants feelings toward trial participation and the outcomes of treatment. Furthermore, our approach to qualitative research mirrored collecting and reflexively analysing in-depth interview data [32], illustrate a process designed to ensure findings adhere to the requirements for study quality in qualitative research as set forth by Lincoln and Denzin [33]. Limitations of this study are low study retention with less than half of the participants unable to be followed up prospectively; shown in Fig 1. Fifteen men were initially interviewed after enrolment but pre-randomisation (commencement of trial). At mid-point (11 weeks) of treatment, three participants were deceased. The study also did not record outcomes regarding men who withdrew from this study and the TheraP trial to subsequently access $^{177}$Lu-PSMA-617 privately. This study also may not be generalisable to other socioeconomic and cultural contexts. Further information regarding ethnicity and socioeconomic makeup of participants could have been identified during the consent and interview process and included in this study. This is an important contextual factor to include in future clinical trials and qualitative studies.

## Conclusion

This study provides insight into the longitudinal experience of men receiving a novel treatment versus chemotherapy in a randomised clinical trial for advanced prostate cancer. Our findings highlight a preference among participants for the novel treatment due to a range of factors. Additionally, our work reveals that in context of limited choices, anticipating outcomes and coping with change, a range of supportive care needs emerged related to physical and mental health symptom management and practical needs associated with their disease, treatment, and trial participation. For both trialists and participants, our findings indicate the need for survivorship care within trials that focus on patient-centred care and personal agency. Patient experiences acquired from qualitative studies could help inform the design and review of future clinical trials.

## Supporting information

**S1 File.**
(DOCX)

**S2 File.**
(DOCX)

## Acknowledgments

The authors would like to thank all participants involved in this study. The QualTheraP sub-study was completed in partnership with the Australian and New Zealand Urogenital and Prostate (ANZUP) Cancer Trials Group and Prostate Cancer Foundation of Australia (PCFA). We would also like to thank the Peter MacCallum Cancer Centre and Royal Brisbane and Women's Hospital for their assistance to the study team. MSH would additional like thank the Prostate Cancer Foundation (PCF) supported by CANICA AS, Oslo, Norway; and the Peter MacCallum Foundation for program support.

## Author Contributions

**Conceptualization:** Suzanne K. Chambers, Nicholas Ralph.

**Formal analysis:** Bianca Viljoen, Nicholas Ralph.

**Funding acquisition:** Michael S. Hofman, Suzanne K. Chambers, Jeff Dunn, Haryana M. Dhillon, Ian D. Davis, Nicholas Ralph.

**Investigation:** Bianca Viljoen.

**Project administration:** Bianca Viljoen, Nicholas Ralph.

**Resources:** Bianca Viljoen, Nicholas Ralph.

**Supervision:** Suzanne K. Chambers, Jeff Dunn, Nicholas Ralph.

**Visualization:** Bianca Viljoen.

**Writing – original draft:** Bianca Viljoen, Nicholas Ralph.

**Writing – review & editing:** Bianca Viljoen, Michael S. Hofman, Suzanne K. Chambers, Jeff Dunn, Haryana M. Dhillon, Ian D. Davis, Nicholas Ralph.

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
