## [Decision Letter · Decision Letter 0]

27 May 2022

PONE-D-22-10351Experiences of participants in a clinical trial of a novel radioactive treatment for advanced prostate cancer: A nested, qualitative longitudinal studyPLOS ONE

Dear Dr. Ralph,

Thank you for submitting your manuscript to PLOS ONE. After careful consideration, we feel that it has merit but does not fully meet PLOS ONE’s publication criteria as it currently stands. Therefore, we invite you to submit a revised version of the manuscript that addresses the points raised during the review process.

In particular please focus on comments raised by reviewers 2 and 3, below.

We look forward to receiving your revised manuscript.

Kind regards,

Randall J. Kimple

Academic Editor

PLOS ONE

Journal Requirements:

[The authors declare there are no competing interests.]

Reviewers' comments:

Reviewer's Responses to Questions

**Comments to the Author**

1. Is the manuscript technically sound, and do the data support the conclusions?

Reviewer #1: Yes

Reviewer #2: Yes

Reviewer #3: Partly

2. Has the statistical analysis been performed appropriately and rigorously? 

Reviewer #1: Yes

Reviewer #2: N/A

Reviewer #3: N/A

3. Have the authors made all data underlying the findings in their manuscript fully available?

Reviewer #1: Yes

Reviewer #2: Yes

Reviewer #3: Yes

4. Is the manuscript presented in an intelligible fashion and written in standard English?

Reviewer #1: Yes

Reviewer #2: Yes

Reviewer #3: Yes

5. Review Comments to the Author

Reviewer #1: Nicely written manuscript reporting on patient experiences as participants in a clinical trial. Important to report on such perspectives especially with novel therapeutic agents, including the radiopharmaceutical agent 177Lu-PSMA-617. Accept as is.

Reviewer #2: The authors report the results of a qualitative study nested within the TheraP clinical trial. The authors should be applauded for this approach and congratulated on this manuscript. There are a few points to consider that may improve the manuscript for readers, and expand upon some of the limitations and implications of this study.

1) Please consider shortening the introduction, making the background information more concise.

2) The potential financial impacts of participating in a trial are not addressed in the introduction, but were included in the interviews.

3) Figure 1 and Table 2: Although the randomization of this trial was 1:1, the participants in this study were 2:1 overrepresented in the Lutetium arm. Please comment on this in the study limitations or elsewhere.

4) Figure 1: Should the arrow for "Refused" go above the "Potential Participants" box?

4) Table 2: Could any data on the ethnic or socioeconomic makeup of the participants be included, beyond education and occupation?

5) Table 3: Under the "Mental changes" section, it appears all the included quotes are from patients who received cabazitaxel. Could any sample quotes from patients who received lutetium be included?

6) Did financial hardship emerge as a theme? It is grouped under the "Practical circumstances" sub-section (line 334). Access to Lutetium PSMA therapy and the financial cost to patients is a significant concern among healthcare providers in the US. It may be worthwhile expanding on this a bit more either within the results or discussion, if the results from the interviews would support such a discussion.

7) Under "Strengths and Limitations" it may be worth further expanding upon the limitation that "this study may not be generalizable to other socioeconomic and cultural contexts". Additional socioeconomic information in table 3 may help provide perspective for this point.

8) Another point to consider including is that patient experiences obtained from qualitative studies could help inform the design and review of future clinical trials.

Reviewer #3: Abstract: Results section is very short. Although this is meant to be a qualitative analysis, it seems that a few summary statistics would be helpful. For example, could the author’s add the percent of men whose experience included each of the 3 ‘identified themes’? Also, could they include some mention of how these evolved over time as mentioned in the objective?

The patients included in this study are a small proportion of the patients included in the larger trial. The conclusions of the paper are targeted to the broader population of patients represented in the larger trial. It would be helpful to include in the results some comparison of characteristics of these 15 patients to all patients on the randomized trial. Are they similar age, … This could be added to Table 2.

6. PLOS authors have the option to publish the peer review history of their article (what does this mean?). If published, this will include your full peer review and any attached files.

Reviewer #1: **Yes: **Freddy E Escorcia

Reviewer #2: No

Reviewer #3: No

---

## [Author Response · Author response to Decision Letter 0]

18 Jul 2022

Response to Reviewer 1

Thank you for your review of our paper. We have answered each of your comments below:

Reviewer Comments Authors Response

Nicely written manuscript reporting on patient experiences as participants in a clinical trial. Important to report on such perspectives especially with novel therapeutic agents, including the radiopharmaceutical agent 177Lu-PSMA-617. Accept as is. Thank you for the comment. 

Response to Reviewer 2

Thank you for your review of our paper. We have answered each of your comments below:

Reviewer Comments Authors Response

The authors report the results of a qualitative study nested within the TheraP clinical trial. The authors should be applauded for this approach and congratulated on this manuscript. There are a few points to consider that may improve the manuscript for readers and expand upon some of the limitations and implications of this study.

1) Please consider shortening the introduction, making the background information more concise.

 Thank you for the comment. 

The authors have reduced the length of the introduction with the aim of making the background information more succinct (Line 74-75 sentences removed, next paragraph moved up from line 78 to 77, line 117- 118, and line 124-126 sentences removed).

2) The potential financial impacts of participating in a trial are not addressed in the introduction but were included in the interviews. The financial impact of participation was only minimally identified by participants in this study. To support the statement in the results (line 339) under the subheading ‘Practical Circumstances’, a comment has been added to line 103 of the introduction. The included references support this. 

3) Figure 1 and Table 2: Although the randomization of this trial was 1:1, the participants in this study were 2:1 overrepresented in the Lutetium arm. Please comment on this in the study limitations or elsewhere. 

This is a reasonable comment however the authors feel that with the variance of four men it is unlikely to be significant for qualitative research outcomes. 

4) Figure 1: Should the arrow for "Refused" go above the "Potential Participants" box? We agree with this comment. Figure 1 has been amended to show that out of the 22 total referrals, 2 refused, resulting in 20 potential and consenting participants. 

Table 2: Could any data on the ethnic or socioeconomic makeup of the participants be included, beyond education and occupation? During the consent and interview process, the ethnicity or socioeconomic make up of participants was not asked or obtained in detail. This data point was not collected as part of the main trial either and has not been standard practice in Australian and New Zealand trials (beyond Indigenous status). This information can therefore not be included in the research paper. We have included a comment in the “Strengths and Limitations” (line 430-432) to address this.

5) Table 3: Under the "Mental changes" section, it appears all the included quotes are from patients who received cabazitaxel. Could any sample quotes from patients who received lutetium be included? Participants who received LuPSMA experienced fewer side-effects and were less anxious about receiving treatment compared to those who received cabazitaxel. Therefore, the only quotes identifying any mental changes from those who received LuPSMA were included in the results under the sub-heading “coping with changes to mental health” (line 303 – 319). A greater number of participants who received cabazitaxel expressed a range of negative emotions. Consequently, more quotes were available and included in the results section and in Table 3. 

6) Did financial hardship emerge as a theme? It is grouped under the "Practical circumstances" sub-section (line 334). Access to Lutetium PSMA therapy and the financial cost to patients is a significant concern among healthcare providers in the US. It may be worthwhile expanding on this a bit more either within the results or discussion, if the results from the interviews would support such a discussion.

Financial impact was not identified strongly enough as a theme. Therefore, it was not included in the discussion. 

All considerable and identified themes are concentrated on and included in the discussion and conclusion. 

7) Under "Strengths and Limitations" it may be worth further expanding upon the limitation that "this study may not be generalizable to other socioeconomic and cultural contexts". Additional socioeconomic information in table 3 may help provide perspective for this point. During the consent and interview process, the ethnicity or socioeconomic make up of participants was not asked or obtained in detail. We have included a comment in the “Strengths and Limitations” (line 430-433) to address this. 

8) Another point to consider including is that patient experiences obtained from qualitative studies could help inform the design and review of future clinical trials. We agree with the reviewers comment. A statement has been added to the end of the Conclusion (Page 23, line 440).

Response to Reviewer 3

Thank you for your review of our paper. We have answered each of your comments below:

Reviewer Comments Authors Response

Abstract: Results section is very short. Although this is meant to be a qualitative analysis, it seems that a few summary statistics would be helpful. For example, could the author’s add the percent of men whose experience included each of the 3 ‘identified themes’? Also, could they include some mention of how these evolved over time as mentioned in the objective? The focus of this study was to describe the lived experiences of men with advanced prostate cancer who participated in the TheraP clinical trial using qualitative analysis. The purpose and execution of qualitative studies is to provide rich data to understand the phenomenon under study; in this instance, clinical trial participation. The authors feel that due to the qualitative design and small number of participants numbers, it does not permit quantitative conclusions. 

The patients included in this study are a small proportion of the patients included in the larger trial. The conclusions of the paper are targeted to the broader population of patients represented in the larger trial. It would be helpful to include in the results some comparison of characteristics of these 15 patients to all patients on the randomized trial. Are they similar age, … This could be added to Table 2. Specific data from participants in the larger trial can unfortunately not be included in the paper for comparison. 

Generalisation of qualitative research via meta-synthesis or narrative in a setting where evidence for improving practice is held in high esteem, warrants careful attention by both qualitative and quantitative researchers (Polit & Beck 2010). Qualitative studies presenting an in-depth understanding of human experience, may also pursue theoretical generalisability (Carminati, 2018).

In regards to the conclusions of the paper, we feel that producing explanations which are particular to the limited empirical parameters of the study is not enough and that our qualitative research should produce explanations which are generalisable, or which have a wider resonance. With the small sample of participants, this has yielded unique insights by revealing symmetries that may be overlooked in large sample studies. As stated in the book ‘Interpreting Qualitative Data’ by David Silverman (2019), good deductions articulate that the data gathered are significant beyond the particular cases, individuals or sites studied.

 

Response from Academic Editor

Thank you for your review of our paper. We have answered each of your comments below:

Editor Comments Authors Response

1. Please ensure that your manuscript meets PLOS ONE’s style requirements, including those for file naming. 

PLOS ONE style requirements have been adhered to. Amendments to the authors affiliations, main body and file naming have been made.

2. Please note that the grant information you provided in the ‘Funding Information’ and ‘Financial Disclosure’ sections do not match. When you resubmit, please ensure that you provide the correct grant numbers for the awards you received for your study in the ‘Funding Information’ section. 

Funding support was received from an ANZUP Below the Belt Research Grant (QualTheraP: A nested, multi-perspective longitudinal qualitative study of participants). There is no associated grant number. 

Interview data, in the form of audio recordings and de-identified transcripts are stored in secure, password protected USQ cloud storage. Only authors on this paper have access to transcript and audio files. Any raw qualitative data will not be made publicly available. Although de-identified it may contain detailed information about a person that could be recognisable to another when read in full.

Data is available on request to authors. 

[The authors declare there are no competing interests.]

 This information should be included in your cover letter; we will change the online submission form on your behalf

The authors have declared that no competing interests exists.

The statement has been included in the cover letter.

---

## [Decision Letter · Decision Letter 1]

17 Aug 2022

PONE-D-22-10351R1Experiences of participants in a clinical trial of a novel radioactive treatment for advanced prostate cancer: A nested, qualitative longitudinal studyPLOS ONE

Dear Dr. Ralph,

Thank you for submitting your manuscript to PLOS ONE. After careful consideration, we feel that it has merit but does not fully meet PLOS ONE’s publication criteria as it currently stands. Therefore, we invite you to submit a revised version of the manuscript that addresses the points raised during the review process.

Please make the minor edits suggested by reviewers 1-3. If you wish to address the comment of reviewer 4 you may do so, but it is not necessary.

We look forward to receiving your revised manuscript.

Kind regards,

Randall J. Kimple

Academic Editor

PLOS ONE

Journal Requirements:

Additional Editor Comments:

Thank you for your patience as this manuscript underwent revision and your clear responses to reviewer comments.

Reviewers' comments:

Reviewer's Responses to Questions

**Comments to the Author**

1. If the authors have adequately addressed your comments raised in a previous round of review and you feel that this manuscript is now acceptable for publication, you may indicate that here to bypass the “Comments to the Author” section, enter your conflict of interest statement in the “Confidential to Editor” section, and submit your "Accept" recommendation.

Reviewer #1: All comments have been addressed

Reviewer #2: All comments have been addressed

Reviewer #3: (No Response)

Reviewer #4: All comments have been addressed

2. Is the manuscript technically sound, and do the data support the conclusions?

Reviewer #1: Yes

Reviewer #2: Yes

Reviewer #3: Partly

Reviewer #4: Partly

3. Has the statistical analysis been performed appropriately and rigorously? 

Reviewer #1: Yes

Reviewer #2: N/A

Reviewer #3: N/A

Reviewer #4: No

4. Have the authors made all data underlying the findings in their manuscript fully available?

Reviewer #1: Yes

Reviewer #2: Yes

Reviewer #3: No

Reviewer #4: No

5. Is the manuscript presented in an intelligible fashion and written in standard English?

Reviewer #1: Yes

Reviewer #2: Yes

Reviewer #3: Yes

Reviewer #4: Yes

6. Review Comments to the Author

Reviewer #1: Manuscript remains a very valuable report on a generally overlooked theme in clinical trials and would be of broad interest to the readership of PLoS One

Minor: Page 5, line 118: Did authors mean “deprivation” instead of “detection”

Reviewer #2: Small editorial correction: Introduction, page 4: "androgen detection therapy" should be "androgen deprivation therapy".

Reviewer #3: I previously suggested that the authors 'include in the results some comparison of characteristics of these 15 patients to all patients on the randomized trial. Are they similar age, … '. The authors have responded that 'Specific data from participants in the larger trial can unfortunately not be included in the paper for comparison.' It is not clear to me why this is? Are you not allowed to give the mean age of all patients enrolled on the larger trial? Is this not already published?

Reviewer #4: This qualitative study describes the lived experiences of men with advanced prostate cancer participating in the TheraP trial; an open-label randomised trial of 177Lu-PSMA-617 compared with cabazitaxel chemotherapy. An interpretative phenomenological approach was used, and interviews analysed using thematic analysis. The current manuscript reports the results from the mid-point, conclusion and follow up interviews, focusing specifically on participants’ experiences of trial participation.

1) The current manuscript identifies three representative themes from the participants. As an open-label trial, it is disappointing the authors didn’t try to identify the differences, if there is any, of the participants between the two arms, which can be very interesting. Since all the materials are available, please dig into it more, and present the findings, even if it’s null.

7. PLOS authors have the option to publish the peer review history of their article (what does this mean?). If published, this will include your full peer review and any attached files.

Reviewer #1: **Yes: **Freddy E Escorcia

Reviewer #2: No

Reviewer #3: No

Reviewer #4: No

---

## [Author Response · Author response to Decision Letter 1]

19 Sep 2022

Comments to the Author

We have answered each of these comments below:

Comments Authors Response

3. Has the statistical analysis been performed appropriately and rigorously?

Reviewer #4: No

We are unsure why there is a comment about the rigour of statistical analyses in a qualitative paper. An interpretative phenomenological approach was used, and interviews analysed using thematic analysis. The current manuscript reports the results from the mid-point, conclusion and follow up interviews, focusing specifically on participants’ experiences of trial participation.

Comments Authors Response

4. Have the authors made all data underlying the findings in their manuscript fully available. The PLOS Data policy requires authors to make all data underlying the findings described in their manuscript fully available without restriction, with rare exception. The data should be provided as part of the manuscript or its supporting information or deposited to a public repository. 

Reviewer #3: No 

Reviewer #4: No 

 All data underlying the findings described in the manuscript have been made available to PLOS ONE as supplementary files. Participants information has been kept confidential. 

Response to Reviewer 1

Thank you for your review of our paper. We have answered your comment below:

Reviewer Comments Authors Response

Minor: Page 5, line 118: Did authors mean “deprivation” instead of “detection”. The word has been amended to state ‘deprivation’. 

 

Response to Reviewer 2

Thank you for your review of our paper. We have answered your comment below:

Reviewer Comments Authors Response

Small editorial correction: Introduction, page 4: “androgen detection therapy” should be “androgen deprivation therapy”. 

 The word has been amended to state ‘deprivation’.

Response to Reviewer 3

Thank you for your review of our paper. We have answered your comment below:

Reviewer Comments Authors Response

I previously suggested that the authors ‘include in the results some comparison of characteristics of these 15 patients to all patients on the randomized trial. Are they similar age, …’. The authors have responded that ‘Specific data from participants in the larger trial can unfortunately not be included in the paper for comparison’. It is not clear to me why this is? Are you not allowed to give the mean age of all patients enrolled on the larger trial? Is this not already published?

The mean (SD) age of participants in the larger TheraP trial have been included in Table 2. Furthermore, the total participants allocated to the LuPSMA and Cabazitaxel arms have also been included, as published in the Lancet: https://www.thelancet.com/journals/lancet/article/PIIS0140-6736(21)00237-3/fulltext

We wish to make the distinction that the language used to draw conclusions does not necessarily relate to the broader population of trial patients in the TheraP trial. 

Response to Reviewer 4

Thank you for your review of our paper. We have answered your comment below:

Reviewer Comments Authors Response

This qualitative study describes the lived experiences of men with advanced prostate cancer participating in the TheraP trial; an open-label randomised trial of 177Lu-PSMA-617 compared with cabazitaxel chemotherapy. An interpretative phenomenological approach was used, and interviews analysed using thematic analysis. The current manuscript reports the results from the mid-point, conclusion and follow up interviews, focusing specifically on participants’ experiences of trial participation.

1) The current manuscript identifies three representative themes from the participants. As an open-label trial, it is disappointing the authors didn’t try to identify the differences, if there is any, of the participants between the two arms, which can be very interesting. Since all the materials are available, please dig into it more, and present the findings, even if it’s null.

The results of the study identify the overall lived experiences of men with advanced prostate cancer participating in the TheraP trial. The authors believe that the similarities and differences in experiences are shown in the results through the inclusion of participants perspectives and selected quotations from participants in both the LuPSMA and Cabazitaxel treatment arms. 

Some changes have been made to the results to try and emphasise the comparison even more (page 14, line 210; page 15 line 232 and 250).

---

## [Decision Letter · Decision Letter 2]

28 Sep 2022

Experiences of participants in a clinical trial of a novel radioactive treatment for advanced prostate cancer: A nested, qualitative longitudinal study

PONE-D-22-10351R2

Dear Dr. Ralph,

We’re pleased to inform you that your manuscript has been judged scientifically suitable for publication and will be formally accepted for publication once it meets all outstanding technical requirements.

Kind regards,

Randall J. Kimple

Academic Editor

PLOS ONE

Additional Editor Comments (optional):

Reviewers' comments:

Reviewer's Responses to Questions

**Comments to the Author**

1. If the authors have adequately addressed your comments raised in a previous round of review and you feel that this manuscript is now acceptable for publication, you may indicate that here to bypass the “Comments to the Author” section, enter your conflict of interest statement in the “Confidential to Editor” section, and submit your "Accept" recommendation.

Reviewer #4: All comments have been addressed

2. Is the manuscript technically sound, and do the data support the conclusions?

Reviewer #4: Yes

3. Has the statistical analysis been performed appropriately and rigorously? 

Reviewer #4: N/A

4. Have the authors made all data underlying the findings in their manuscript fully available?

Reviewer #4: Yes

5. Is the manuscript presented in an intelligible fashion and written in standard English?

Reviewer #4: Yes

6. Review Comments to the Author

Reviewer #4: My comments have been addressed. I have no further critique.

My comments have been addressed. I have no further critique.

7. PLOS authors have the option to publish the peer review history of their article (what does this mean?). If published, this will include your full peer review and any attached files.

Reviewer #4: No

---

## [Editor Report · Acceptance letter]

14 Oct 2022

PONE-D-22-10351R2 

Experiences of participants in a clinical trial of a novel radioactive treatment for advanced prostate cancer: A nested, qualitative longitudinal study 

Dear Dr. Ralph:

I'm pleased to inform you that your manuscript has been deemed suitable for publication in PLOS ONE. Congratulations! Your manuscript is now with our production department. 

Kind regards, 

on behalf of

Dr. Randall J. Kimple 

Academic Editor

PLOS ONE